Bird-window collisions in the summer breeding season

Hager Stephen B. 1 stevehager@augustana.edu
Craig Matthew E. 1 2
1 Department of Biology, Augustana College , Rock Island, IL , USA
Kass Philip
2 Current affiliation: Department of Biology, Indiana University, Bloomington, IN, USA

Electronic publication date: 2014 Jun 26
Publication date: 2014
Volume: 2
Electronic Location ID: e460
Received 2014 May 8; Accepted 2014 Jun 8
Copyright: © 2014 Hager and Craig
Copyright year: 2014
Copyright holder: Hager and Craig
License: This is an open access article distributed under the terms of the Creative Commons Attribution License, which permits unrestricted use, distribution, reproduction and adaptation in any medium and for any purpose provided that it is properly attributed. For attribution, the original author(s), title, publication source (PeerJ) and either DOI or URL of the article must be cited.
License URL: https://creativecommons.org/licenses/by/4.0/

Keywords: Human threats, Bird-window collisions, Breeding birds, Urban ecology, Avian mortality, Window collision, Buildings, Glass

Funding: Summer Research Fellowship at Augustana College MEC received funding through a Summer Research Fellowship at Augustana College. The funders had no role in study design, data collection and analysis, decision to publish, or preparation of the manuscript.

==============================
Birds that reside in urban settings face numerous human-related threats to survival, including mortality from bird-window collisions (BWCs). Our current understanding of this issue has largely been driven by data collected during spring and fall migration, and patterns of collision mortality during the summer breeding season remain relatively unexplored. We assessed BWCs during four breeding seasons (2009–2012) at a site in northwestern Illinois, USA, by comparing the abundance, richness, migratory class, and age of the species living around buildings to species mortally wounded by window collisions. We also systematically assessed the daily timing of BWCs throughout the breeding season. We documented BWCs in 4 of 25 (16%) species and 7 of 21 (33%) species in 2009 and 2010, respectively. The relationship between BWCs and abundance depended on age. For adults, BWCs were highest in the least abundant species, e.g., Red-eyed Vireo (Vireo olivaceus), and lowest in species with high abundance values, e.g., Chipping Sparrow (Spizella passerina). For juveniles, mortality was greatest for the most abundant species, and the American Robin (Turdus migratorius) accounted for 62% of all juvenile carcasses. Early in the breeding season, collision mortality was restricted to adults of Long-distance Migrants, whereas juveniles of all three migratory guilds (Long-distance and Short-distance Migrants and Permanent Residents) died at windows from late June through early August. Daily mortality for all species was highest between sunrise–1600 h and lowest from 1600 h–sunrise the next day. Generally, the species observed as carcasses matched birds considered a ‘high risk’ for BWCs, e.g., Ruby-throated Hummingbird (Archilochus colubris), and those considered ‘low risk’ were not observed as carcasses, e.g., Blue-gray Gnatcatcher (Polioptila caerulea). Our results suggest that the number of BWCs during the breeding season does not necessarily increase with abundance, but rather appears related to variation among species and age classes, which may have important implications on the population health of affected species. The mechanisms driving these differences are unknown, but may be related reproductive behavior, flight speed, distance movements, and dispersal patterns.

Introduction

Birds that reside in urban settings face numerous human-related threats to survival, including mortality from bird-window collisions (‘BWCs’; Fig. 1A). Knowledge of the drivers of BWCs is necessary to effectively mitigate the impacts of BWCs. Previous work demonstrates significant spatial variation in bird mortality resulting from window strikes. In particular, buildings with high window area and surrounding green space incur the greatest frequency of BWCs and are patchily distributed across the urban landscape (Bayne, Scobie & Rawson-Clark, 2012; Hager et al., 2013; Machtans, Wedeles & Bayne, 2013; Loss et al., 2014). However, despite knowledge that bird behaviors differ across seasons, days, and species, we have an incomplete understanding of temporal and species-specific patterns of BWCs. This information is necessary to inform full life cycle population modeling and population health assessments, which would assist in conservation efforts aimed at reducing collision-related impacts (Loss, Will & Marra, 2012).

Figure 1 Birds fatally wounded after crashing into windows.

(A) Hermit Thrush (Catharus guttatus) fatally wounded after crashing into a window. (B) Feather pile of a Gray Catbird (Dumetella carolinensis) carcass, which resulted from a window collision. Feather piles are produced when decomposers and vertebrate scavengers, such as raccoons (Procyon lotor), remove most soft and bony tissue, and leave behind various feathers (e.g., remiges, nonflight feathers, and rectrices), body parts (e.g., wing, tail, and legs), and soft tissue (e.g., intestines) (Hager, Cosentino & McKay, 2012). Bird identification is possible if species-specific feathers are present. Moreover, feather piles remain detectable by field workers for several days, and thus, provide evidence of collisions (see Results, Scavenger Assessment).

Collision mortality is reported to be highest during spring and fall migration relative to bird residency during winter and summer (Drewitt & Langston, 2008). However, this conclusion has been derived from studies that were conducted mainly during migration (Drewitt & Langston, 2008). A recent experiment employing a systematic sampling design for all seasons confirmed low mortality in winter, but suggested that collisions during the breeding season, i.e., June-early August, are similar in magnitude to spring and fall migration (Hager et al., 2013). In addition to variability across seasons, BWCs likely vary within a 24-h period due high levels of morning activity related to feeding and behavioral interactions within and among species (McNamara, Mace & Houston, 1987). In the breeding season, daily activities patterns would also be affected by high ambient temperatures resulting in low levels of activity during hot afternoons, and reproductive behaviors, such as dispersal of postfledging individuals (Whittaker & Marzluff, 2012), which may further influence risk of window strikes. A better understanding of the temporal patterns of mortality within the breeding season would clarify our current view of the threat posed by BWCs.

In addition to potential temporal dynamics of BWCs during the breeding season, the resident community during these months may determine the species, migratory guild, and ages of affected individuals. Short and Long-distance migrants, including birds of conservation concern, and hatch-year birds appear to experience the highest incidence of mortality (Hager et al., 2013; Loss et al., 2014). Moreover, it has been asserted that the abundance and richness of birds are the best predictors of BWCs (Klem, 1989). However, these conclusions deserve further scrutiny because, as stated earlier, little work has focusesd on mortality outside of spring and fall migration (Drewitt & Langston, 2008). Breeding bird communities may be composed of Short- and Long-distance Migrants, Permanent Resident species, and both adults and post-fledgling individuals (Blair, 1996). Thus, assessing whether BWCs are related to a species, migration strategy, age class, and abundance is ideally suited to the summer breeding season.

We addressed the need for a better understanding of summertime BWCs by documenting the temporal dynamics of and species affected by BWCs for a breeding bird community in northwestern Illinois, USA. This was evaluated during four summer seasons on a college campus composed of low-rise commercial buildings situated in moderate to high levels of green space. In 2009–2010, we used point count surveys to estimate the abundance, richness, migratory guild, and age class of the site’s breeding birds, and compared these metrics to the species mortally wounded by window collisions. In 2011–2012, we completed systematic surveys at five intervals each day to assess how mortality was distributed throughout a 24-h period. Birds affected by window collisions in our study were then compared to vulnerability estimates for species in the United States (Loss et al., 2014). Bias associated with imperfect detection of carcasses was reduced in all summers by accounting for the effects of scavengers and carcass detection by field workers.

Materials and Methods

Study site

We assessed BWCs at Augustana College in northwestern Illinois, USA, for nine weeks of each breeding season (June–early August) from 2009 to 2012. The campus was constructed on 0.65 km2 within the bluffs of the Mississippi River (90°33′W, 41°30′N) and located in the Eastern Tallgrass Prairie Bird Conservation Region (Sauer, Fallon & Johnson, 2003). Bluff faces contained moderately disturbed deciduous hardwood forest (‘wooded bluff faces’), and terraces above and below bluffs were similar in structure to grassland savanna with scattered woody trees and shrubs and an open understory of landscaped grass (‘landscaped savanna’). Work was completed at four low-rise commercial buildings: Westerlin Residence Hall (height =3 stories, building footprint = 5,690 m2), Hanson Hall of Science (height = 5 stories, building footprint = 2,432 m2), Thomas Tredway Library (height = 5 stories, building footprint = 1,837 m2), and Swenson Hall of Geosciences (height = 3 stories, building footprint = 745 m2).

Carcass surveys

In 2009 and 2010, we completed carcass surveys at intervals of 1–3 days at Hanson Hall of Science, Swenson Hall of Geosciences, and Thomas Tredway Library. During each survey, a trained fieldworker walked a complete transect around a building’s footprint within a 2-m buffer. This buffer accommodated the perpendicular distance from external walls at which most carcasses are located (95% CI [93–127 cm]; N = 51 carcasses; S Hager, 2013, unpublished data). A bird carcass consisted of a full body, partial carcass, or feather pile (Hager & Cosentino, 2014). We assumed high average detection probability of bird carcasses (0.88, SE = 0.01) based on Hager et al. (2013).

In 2011 and 2012, we assessed the daily pattern of BWCs by surveying for carcasses for four consecutive days per week at Hanson Hall of Science, Swenson Hall of Geosciences, and Westerlin Residence Hall, which was included because of construction activities that prevented access to Thomas Tredway Library. Within each day-building combination, five surveys were completed at sunrise, 0900 h (CDT), 1230 h, 1600 h, and one hour prior to sunset. We assumed that a carcass found during a survey died in the interval between that time and the previous survey. A ‘clean-up survey’ was conducted at sunset the day before the first sunrise survey for each sampling week (Hager & Cosentino, 2014). This survey removed all carcasses that accumulated between survey-weeks, which may have otherwise introduced detection bias on the first day of weekly sampling.

Carcasses and corresponding identification tags were placed in zip-lock plastic bags and later identified to species in the laboratory. Birds were classified as adult or juvenile based on plumage, degree of cranial pneumatization, and, in hummingbirds, pattern of bill serration (Pyle, 1997). We consulted Fair et al. (2010) for recommendations related to collecting procedures of bird carcasses. Carcasses were salvaged under state Scientific Permit (NH11.0313), Illinois Department of Natural Resources, and federal Salvage Permit (MB08907A-0), U.S. Fish and Wildlife Service.

Point count surveys

We used point counts to rapidly assess the population sizes of all species present on site in 2009 and 2010 (Johnson, 2008). Our goal for point count information was to describe the community of breeders in close proximity to study buildings (Hager et al., 2013). Thus, we established one 50-m radius count circle each in wooded bluff face and landscaped savanna, and count circle edges were within 50–125 m of each of study buildings. We reasoned that this information would most closely approximate the number and types of birds at risk of hitting windows at the buildings we were monitoring. Two surveys/week in June and July were completed during appropriate weather conditions and within 4 h after sunrise (Bled et al., 2013).

We identified and counted all birds seen and heard within a 5-min count period. Species richness was the total number of species observed, and abundance was the maximum number of individuals for each species, which is an appropriate community metric for repeated counts conducted at the same point location (Johnson, 2008). We assumed that the maximum abundance for each species was a reflection of the abundance of both adults and juveniles because we could not distinguish birds of various age classes during count surveys. The following species, guilds, and taxonomic groups were excluded from analyses: birds flying over the site, migratory flocks, waterfowl, and raptors (Kalinowski & Johnson, 2010; Hager et al., 2013). We followed the recommendations of Fair et al. (2010) in reducing impacts to birds resulting from investigator presence during point count surveys. All point count surveys were completed by SBH to reduce detection biases that may result from multiple counters.

We used North American Ornithological Atlas Committee (2012) to classify levels of each species’ breeding behaviors, which were documented opportunistically during point counts and at other times throughout the season. Species classified above the ‘observed’ level were considered part of the site’s breeding community.

Scavenger study

We minimized bias associated with imperfect detection of carcasses by using a sampling design that was informed by estimates of carcass persistence before removal by scavengers and decomposers and detection by field workers (Hager & Cosentino, 2014). Persistence or survival of whole and partial carcasses, i.e., feather piles (Fig. 1B), was monitored at four campus buildings (Hanson Hall of Science, Swenson Hall of Geosciences, Thomas Tredway Library, and Olin Center for Education Technology) during a 7-d study period (2–8 June 2010). Two of eight whole, intact bird carcasses were randomly placed below different facades of each building (see Hager, Cosentino & McKay, 2012 for details on carcass placement and daily monitoring). Each carcass was a different species ranging in size from Tennessee Warbler (Oreothlypis peregrina; 9 g, Sealy, 1985) to Brown Thrasher (Toxostoma rufum; 70 g, Cavitt & Haas, 2000).

Data analysis

For the carcass detection study, we estimated mean survival times (t) for complete and partial (i.e., feather piles) carcasses at each building using the exponential model r = e−d/t, where r is the probability of survival for d = 1 day (Huso, 2011; Hager, Cosentino & McKay, 2012).

We used a Fisher’s Exact Test to examine differences in the number of BWCs among species considered Short-distance Migrant, Long-distance Migrant, and Permanent Resident.

To determine whether species abundance was related to the number of collisions, we used a generalized linear model, specifying a Poisson distribution with a log link function. We included maximum abundance, carcass age (juvenile or adult), the interaction between carcass age and maximum abundance, and year as predictors of carcass counts, but did not retain year as it was not significant.

We used a general linear model to evaluate the relationship between BWCs and time-of-day. Because inclement weather prevented some surveys from being conducted and because survey intervals were not equivalent, carcass data were standardized and are reported as the number carcasses survey-1 hour-1. Survey time and year were included as predictors of the number of carcasses detected per survey; year was not significant and was removed from the final model.

Results

Scavenger assessment

During the 7-day scavenger study, the probability of daily carcass survival was high (0.8) and average carcass persistence at buildings was 6.25 days. A scavenger removed one carcass within 24 h after initial placement. Within 5 days, decomposers gradually transformed the remaining seven carcasses into feather piles, which were detectable and persisted until the end of the study (Fig. 1B).

Bird density and mortality

In 2009, we documented BWCs in 4 of 25 (16%) species of the breeding community, which we determined using the breeding behavior criteria established by North American Ornithological Atlas Committee (2012) (Fig. 2A). Two species not observed during point counts, and thus not considered part of the breeding community, were collected as carcasses: Belted Kingfisher (Megaceryle alcyon) and Yellow-billed Cuckoo (Coccyzus americanus). In 2010, we documented BWCs in 7 of 21 (33%) breeding species (Fig. 2B). An additional three species were collected as carcasses, but were not considered part of the breeding community, including Ruby-throated Hummingbird (Archilochus colubris), Baltimore Oriole (Icterus galbula), and Indigo Bunting (Passerina cyanea). There was no significant difference in the number of carcass species vs. non-carcass species (N = carcass, non-carcass) among Short-distance Migrants (1, 9), Long-distance Migrants (4, 3), and Permanent Residents (4, 6) (Fisher’s Exact Test, P = 0.15).

Figure 2 Abundance of avian breeding species and those species affected by window collisions.

Maximum abundance of live birds observed during point counts and number of carcasses (adults, juvenile, or could not be aged) resulting from window collisions in northwestern Illinois, USA, in two summer breeding seasons: (A) 2009 and (B) 2010. Species with maximum abundances of zero for a given year were not considered as being part of the breeding community for that year.

We determined the age classes of 24 carcasses, which were 67% juveniles and 33% adults (Fig. 3). Adults (all Long-distance Migrants) were affected by window collisions only at the beginning of the breeding season (Fig. 3A); species (N) included Yellow-billed Cuckoo (1), Ruby-throated Hummingbird (2), Eastern Wood-Pewee, Contopus virens (1), Red-eyed Vireo, Vireo olivaceus (1), Gray Catbird, Dumetella carolinensis (1), Indigo Bunting (1), and Baltimore Oriole (1). Juveniles were documented from the third week in June to the end of July (Fig. 3B), and were principally composed of Short-distance Migrants (N = 10; all American Robins) followed by Long-distance Migrants (N = 2 Ruby-throated Hummingbirds and N = 1 Gray Catbird) and Permanent Residents (N = 1 each of Black-capped Chickadee, Poecile atricapillus, Northern Cardinal, Cardinalis cardinalis, and House Finch, Carpodacus mexicanus).

Figure 3 Timing of bird-window collisions throughout the breeding season.

Migration guilds and number of (A) adult and (B) juvenile birds collected as carcasses resulting from window collisions in northwestern Illinois, USA, for each of nine weeks of the breeding seasons of 2009 and 2010.

Figure 4 Daily timing of bird-window collisions during the breeding season.

Frequency of bird-window collisions throughout a 24-h period during the 2011 and 2012 summer breeding seasons in northwestern Illinois, USA. Solid line represents the quadratic relationship between survey time and the frequency of bird-window collisions in the preceding interval (y = −2 × 10−8x2 + 4 × 10−5x−0.0087).

Overall, the model containing maximum abundance, carcass age, and the interaction between carcass age and maximum abundance explained significant variation in bird mortality (χ2 = 36.3, df = 3, P < 0.001, McFadden’s r2 = 0.26). The relationship between BWCs and abundance depended on age (interaction: χ2 = 12.7, df = 1, P < 0.001, Fig. 2). For adults, bird abundance was correlated negatively with BWCs (β = −0.80 ± 0.22). That is, BWCs were highest in the least abundant species, e.g., Red-eyed Vireo and Eastern Wood-Pewee, and lowest in species observed with high abundance values, e.g., House Sparrow (Passer domesticus), Chipping Sparrow (Spizella passerina), and American Goldfinch (Spinus tristis). For juveniles, mortality was positively related to abundance (β = 0.27 ± 0.05), which was mainly driven by the American Robin (Fig. 2).

Daily patterns of mortality

There was a quadratic relationship between the frequency of BWCs and survey time (r2 = 0.48, F1,7 = 3.2, P = 0.10, Fig. 4). Although this relationship was not significant at α = 0.05, we interpret the model fit as marginally significant at α = 0.1 because our likelihood of detecting a signifcant effect was limited by a small sample size of carcasses (N = 10). Sixty-six percent of birds died between sunrise and 1600 h and the remaining mortality occurred before sunrise. No carcasses were observed between 1600 h and sunset. Carcass species documented in 2011 and 2012 included: American Robin, Ruby-throated Hummingbird, Black-capped Chickadee, and four carcass species that were not observed in 2009 and 2010: Norther Flicker (Colaptes auratus), Ovenbird (Seiurus aurocapilla), White-breasted Nuthatch (Sitta carolinensis), and House Sparrow (Passer domesticus).

Species vulnerability

Eleven of 17 (65%) carcass species observed in all four breeding seasons (2009–2012) were recently listed by Loss et al. (2014) as being highly vulnerable to striking windows in the United States (Table 1). Of these, we found that Ruby-throated Hummingbirds and Gray Catbirds died in the greatest numbers and in at least two of the breeding seasons. However, two species of the breeding community, Cedar Waxwing (Bombycilla cedrorum) and Blue Jay (Cyanocitta cristata), considered vulnerable to BWCs were never found as carcasses. Generally, bird groups estimated to be at high risk and low risk for window collisions corresponded to the bird groups we observed as carcasses (Table 2). For example, Hummingbirds and Swifts and Kingfishers are listed as high-risk groups, which we documented with carcasses from Ruby-throated Hummingbird and Belted Kingfisher. Inconsistent with published vulnerability estimates (Loss et al., 2014) were observations of no mortality in high-risk groups, e.g., Waxwings, and documented mortality in low-risk groups, e.g., Flycatchers and Vireos.

Table 1 Breeding bird species and their vulnerability to window collisions.

Number of carcasses and number of breeding seasons in which carcasses were found of species estimated as being highly vulnerable to window collisions. Data collected from 2009 to 2012 in northwestern Illinois, USA.

Highly vulnerable speciesa	# Carcasses	# Breeding seasons	
Ruby-throated Hummingbird	6	3	
Gray Catbird	3	2	
House Finch	1	1	
Northern Cardinal	1	1	
Downy Woodpecker	1	1	
Black-capped Chickadee	1	1	
Northern Flickerb	1	1	
White-breasted Nuthatchb	1	1	
Ovenbirdb	1	1	
Cedar Waxwing	0	0	
Blue Jay	0	0	
Notes.

a Based on Table 4 in Loss et al. (2014).

b Carcass found in either 2011 or 2012 when estimating daily mortality, but not during the summers of 2009 and 2010 when community composition was assessed.

Table 2 Breeding bird groups and their vulnerability to window collisions.

Comparison between building collision vulnerability for bird groups and species within respective groups that were documented or not documented as carcasses during the breeding seasons 2009–2012, northwestern Illinois, USA. Vulnerability estimates based on Loss et al. (2014). “Risk values indicate the factor by which a species has a greater chance (for positive residuals) or a smaller chance (for negative residuals) of mortality compared with a species with average risk” (Table 5, Loss et al., 2014).

Bird group vulnerability	Residual	Risk	Species found as carcasses	Species not found as carcasses	
Hummingbirds and swifts	1.52	33.2	Ruby-throated Hummingbird	Chimney Swift	
Kingfishers	0.56	3.6	Belted Kingfisher	–	
Waxwings	0.55	3.6	–	Cedar Waxwing	
Warblers	0.54	3.4	Ovenbirda	–	
Nuthatches, tits, and creeper	0.50	3.1	Black-capped Chickadee,
White-breasted Nuthatcha	–	
Cuckoos	0.46	2.9	Yellow-billed Cuckoo	–	
Mimic Thrushes	0.41	2.6	Gray Catbird	–	
Cardinaline Finches	0.36	2.3	Indigo Bunting, Northern Cardinal	–	
Thrushes	0.25	1.8	American Robin	–	
Cardueline Finches	0.23	1.7	House Finch	American Goldfinch	
Woodpeckers	0.15	1.4	Downy Woodpecker,
Northern Flickera	Red-bellied Woodpecker	
Doves and pigeons	0.08	1.2	–	Mourning Dove, Rock Pigeon	
Sparrows	0.08	1.2	–	Chipping Sparrow	
House Sparrow	−0.15	1.4	House Sparrowa	–	
Wrens	−0.20	1.6	–	House Wren, Carolina Wren	
Flycatchers	−0.41	2.6	Eastern Wood-Pewee	Eastern Phoebe	
Vireos	−0.55	3.6	Red-eyed Vireo	–	
Starling	−0.56	3.6	–	European Starling	
Blackbirds, meadowlarks, and orioles	−0.64	4.4	Baltimore Oriole	Common Grackle, Brown-headed Cowbird	
Gnatcatchers	−1.68	48.1	–	Blue-gray Gnatcatcher	
Notes.

a Carcass found in either 2011 or 2012 when estimating daily mortality, but not during the summers of 2009 and 2010 when community composition was assessed.

Discussion

To better understand summer-time BWCs, we used a systematic sampling protocol to assess whether abundance, richness, migratory guild, and age class of a breeding community in Illinois influenced which species were affected by BWCs. In addition, we assessed how mortality varied throughout a 24 h period within breeding seasons. In the scavenger study, carcasses persisted for over 6 days, which when combined with high searcher detection probability (Hager et al., 2013), reduced bias associated with imperfect detection of carcasses.

We found that collision mortality for adults was inversely related to species abundance, and only adult Long-distance Migrants died early in the breeding season. Conversely, juvenile mortality was positively related to species abundance, and juveniles of all three migratory guilds (Long-distance and Short-distance Migrants and Permanent Residents) died from late June through early August. Mortality differences among age classes would be expected to reflect the timing at which individuals were present on site: adult mortality prior to successful reproduction and post-fledging mortality after juveniles enter the population. However, adults generally remain within breeding territories near buildings throughout the entire season and, if collision risk is simply related to abundance, then we should have observed adult mortality more consistently throughout the summer. The mechanism driving mortality differences in age and migratory class is unknown, and because we could not differentiate age classes during point counts, we are not certain whether differences in species or age class abundance drove differences in mortality.

BWCs are hypothesized to be influenced by flight behavior and temporal variation in mobility, i.e., flight speed, distance moved, and dispersal patterns (Klem, 1989). From the perspective of adults, risk of hitting windows may be highest early in the breeding season as individuals engage in high velocity social interactions, such as chases, that are used, among other behaviors, for territory establishment and defense. Following territory settlement, reproductive behavior transitions to brooding of eggs and nestlings resulting in reduced mobility and a decrease in collision risk. Generally, the start of the breeding season in the upper Midwest is staggered among the different migratory guilds. Territorial behavior for Permanent Residents, such as the Downy Woodpecker, and Short-distance Migrants, such as the American Robin, generally begins in February and April, respectively. Thus, the time frame of our fieldwork failed to capture intense territorial behaviors for Permanent Residents and Short-distance Migrants, and instead coincided with reduced mobility and no collisions. In contrast, adults of Long-distance Migrants, such as the Ruby-throated Hummingbird, would have been engaging in aggressive, territorial behavior (Weidensaul et al., 2013) at the start of our summer field seasons, i.e., early June, and therefore observations of relatively high mortality.

Differences in interspecific collision mortality among juveniles may be related to post-fledging dispersal movements, which varies among migratory guilds, foraging requirements, and habitat preferences (Whittaker & Marzluff, 2012; Ausprey & Rodewald, 2013). For example, Whittaker & Marzluff (2012) found that high speed and long distance dispersal movements were associated with migrating species, such as the American Robin, and selection for highly mobile individuals may be a response to ephemeral food sources, i.e., fruiting trees and shrubs and invertebrate concentration. Thus, juvenile robins with high levels of dispersal mobility should die at high rates, which is what we observed. In contrast, previous work has also demonstrated that juveniles of resident grainivorous species have low levels of dispersal mobility and are constrained to residential patches with bird feeders and preferred habitat (Whittaker & Marzluff, 2012; Ausprey & Rodewald, 2013). Indeed, we observed little to no mortality in juveniles of seed eating resident (as well as migrant) species, i.e., juveniles with lower expected dispersal mobility.

We found that the daily timing of collision mortality in the summer was highest between sunrise and 1600 h. These results are generally consistent with Klem (1989) who reported daily mortality at two houses and for all seasons, combined. Summertime mortality may be correlated with particular bird activities throughout the day (e.g., foraging), and infrequent collisions in the late afternoon through sunrise the next day could reflect periods of inactivity when birds are behaviorally thermoregulating due to high ambient temperatures and roosting at night (Robbins, 1981). Although several bird carcasses were assigned to the interval between sunset–sunrise, we didn’t conduct carcass surveys at night, and thus are not confident about whether birds died at the night or in the pre-dawn crepuscular hours.

Overall, our findings document variation in the number of carcasses resulting from window collisions between adults and juveniles. Specifically, risk of BWCs was high in two groups: (1) adults of the least abundant species and (2) juveniles of the most abundant species. We view these results as preliminary because of low replication of study buildings and point count sites, both of which place limitations on inferences beyond the local scale. Moreover, bird detection during point counts varies among species (Johnson, 2008), which may have biased our estimates of richness and abundance for birds with low detection probability, such as Ruby-throated Hummingbird. Despite these limitations, our results support the hypothesis that risk of BWCs varies among species, and suggests that collision mortality during the breeding season may be significant.

Conservation implications and future research

Generally, the carcass species we found conformed to the species and species groups considered to be highly vulnerable to BWCs, such as Ruby-throated Hummingbird and Nuthatches, tits, and chickadees (e.g., Black-capped Chickadee) (Loss et al., 2014). Moreover, we observed limited mortality in birds whose species groups are considered to be at low risk of collisions, e.g., Orioles (Baltimore Oriole), Vireos (Red-eyed Vireo), and Flycatchers (Eastern Wood-Pewee). However, we found no carcasses of some high-risk species, such as Cedar Waxwings.

A more comprehensive understanding of the magnitude of mortality for species and ages affected in this region should be derived from studies that systematically sample a large number of study buildings and that employ a standardized carcass survey protocol (Hager & Cosentino, 2014). Standardized mortality data would allow for direct comparisons among sites and, collectively, to local population estimates from, for example, the North American Breeding Bird Survey (Loss, Will & Marra, 2012). Moreover, these data can be incorporated into models of survival for adults and post-fledglings, which when combined with estimates of nest productivity, would shed important light on population trajectories (Balogh, Ryder & Marra, 2011).

At broad scales, the nature of bird communities among urban areas should reflect landscape structure and functional connectivity (Chace & Walsh, 2006; Ramalho & Hobbs, 2012), which would result in variation in the species and magnitude of BWCs. For example, the American Robin (an urban adapted species) responds positively to urbanization throughout much of its range including desert scrub, closed canopy forest, and grasslands (Blair, 2004; Chace & Walsh, 2006), and high levels of juvenile mortality documented in this study may be occurring in urban landscapes throughout the range of this species. However, differences in avian communities among the major habitat types (i.e., desert scrub, closed canopy forest, and grasslands) should yield unique suites of urban sensitive species that may be vulnerable to window collisions. Future work should assess variation in species affected by BWCs for breeding bird communities across multiple spatial scales, which could then inform studies on demography and population health of those species.

Supplemental Information

Supplemental Information 1 Point count data for each count circle and NORAC Breeding Code for each species observed in 2009 and 2010.

Click here for additional data file.

Supplemental Information 2 Species, ages, and migratory guild of carcasses documented at study buildings in 2009 and 2010.

Click here for additional data file.

Supplemental Information 3 Daily timing of mortality and carcass species observed in 2011 (N = 27 carcass surveys) and 2012 (N = 30 carcass surveys).

Click here for additional data file.

Supplemental Information 4 Carcass and feather pile persistence during a 7-day scavenger assessment in 2010.

Click here for additional data file.

We thank Dan Meden, Conrad Newell, Logan Cygan, Michael Dickens, Andrew Kreiser, and Rachel Mozwecz for their help in conducting carcass surveys. William Phipps, one anonymous reviewer, and the students of Spatial Ecology of Birds in Urban Landscapes (BIOL410) at Augustana College offered helpful comments on an early draft of the manuscript.

Additional Information and Declarations

Competing Interests

Author Contributions

Animal Ethics

Field Study Permissions

The author declares there are no competing interests.

Stephen B. Hager conceived and designed the experiments, performed the experiments, analyzed the data, contributed reagents/materials/analysis tools, wrote the paper, prepared figures and/or tables, reviewed drafts of the paper, organized and managed the assessment of daily bird mortality in 2011 and 2012.

Matthew E. Craig conceived and designed the experiments, performed the experiments, analyzed the data, contributed reagents/materials/analysis tools, wrote the paper, prepared figures and/or tables, reviewed drafts of the paper.

The following information was supplied relating to ethical approvals (i.e., approving body and any reference numbers):

We followed the recommendations of Fair et al. (2010) in reducing impacts to birds resulting from investigator presence during point count surveys. We consulted Fair et al. (2010) for recommendations related to collecting procedures of bird carcasses.

The following information was supplied relating to field study approvals (i.e., approving body and any reference numbers):

Carcasses collected during field surveys and those used in scavenger assessments were salvaged under state Scientific Permit (NH11.0313), Illinois Department of Natural Resources, and federal Salvage Permit (MB08907A-0), U.S. Fish and Wildlife Service.

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
