# Peer review of "Bird-window collisions in the summer breeding season"

_PeerJ, doi:10.7717/peerj.460_

## Round 0.1 · original submission · Minor Revisions

Thank you for submitting your manuscript for publication. It has been reviewed by two experts in this field, and while both were supportive of publication, they have raised a number of points that will need to be addressed point-by-point in a rebuttal letter. If you disagree with the reviewers, please explain carefully why, and your revision may then need to undergo another round of review.

Reviewer 1 ·

Basic reporting

• Abstract: The abstract seems too long. Please reduce.
• Overall the writing seems a bit rambling at times, especially in the Discussion. Please stick to the pertinent information. For example, in the Discussion you write “Unfortunately, one field worker could not start all building surveys at sunrise…” This sentence is unnecessary.
• Figure 1 is unnecessary.

Experimental design

• This study was conducted at only one site, and the sample sizes (carcasses for scavenging study, carcasses found) are low. I still think useful information can be gained in such a situation, but the authors should add a paragraph to the Discussion that covers the limitation to inference caused by this type of local study.
• Scavenger study: Given that only eight carcasses were used, I don’t think you obtained much useful information on potential loss to scavengers. I would delete this section and instead rely on the previous study (Hager et al. 2012).

Validity of the findings

• Data analysis: I am bothered by the fact that you omitted American Robin from your analysis for no good biological reason. It seems that with its inclusion you didn’t get the result you expected or one that you could interpret, so you arbitrarily chose to remove it. If you can give a biological reason why you censured this species then that is fine, but it is not good statistical practice to omit data just because it doesn’t fit the expectation.
• Along those lines, the Figures seem to show that without AMRO included, the reported effect of bird abundance being negatively correlated with BWC seems very weak. You give a P value, but no other statistics in the Results section. What is the strength of the relationship (R-squared, etc.)? This is an important point.
• In Figure 4, there is a trendline (regression line) included. What does this mean? What is the quantitative relationship? Also, you seem to be inferring a linear relationship between # carcasses/survey and the mortality time frame, but the five categories for the mortality time frame aren’t equal. For example, sunset-sunrise must be a longer time period than the others, so how can this be comparable in a linear fashion?

Additional comments

I think there is some good information in this paper (especially the temporal aspects of BWC across the day), but there are several important issues with the analysis that should be remedied before the paper is further considered for publication.

·

Basic reporting

The authors investigated bird-window collisions (BWCs) at a single study site over four breeding seasons and found that species abundance of bird carcasses from BWCs was negatively related to the abundance of breeders (based on point count estimates). The authors also found a difference in the age and migratory class of BWC carcasses depending on the stage of the breeding season, as well as a temporal pattern of BWC risk (with highest risk of BWC at night and in the morning).
Overall this is a well written article that addresses a contemporary research topic of conservation importance. I do have some questions about the methods employed and the subsequent interpretation of the findings, which are detailed below and highlighted on the attached annotated pdf.

1. The Abstract is rather long, with too much detail relating to methodology. There is a danger of the main point of the article being lost. Please refer to highlighted text for sentences that could potentially be removed.

2. The introduction is well written and includes sufficient background information and reference to relevant studies.

3. Figures and tables are acceptable.
I have one query about Figure 2B: The first line of the “Bird density and mortality” section of the results states that 9 species were documented in BWCs. Figure 2B shows 10 species.
Additional explanation would also be useful on the Fig. 4 legend.

4. The “Scavenger study” aspect of the methods section could be moved below the “Point count surveys” section, or even incorporated into the “Carcass surveys” section as it is not the most important aspect of the article.

Experimental design

Although I am generally satisfied with the scavenger study and carcass survey methodologies, I have some concerns about the point count surveys and overall experimental design, detailed below.
1. I understand the logistical constraints of only being able to survey at one study site. However, it should perhaps be mentioned in the text that as the study is only conducted at one site then the findings should be considered as a first insight.

2. The 5 minute duration of the point count seems too short to allow for a sufficient proportion of the breeding birds in each habitat to be detected, particularly given that only one point count site was selected for each of the two dominant habitats. This raises the question of whether the abundance estimates are truly representative of the local avian community, particularly given the high variability in detectability between different bird species.

3. Related to point 2: Were the point count sites located in the same place in both habitats each time? If so then this was perhaps not a representative sample of the whole area of each habitat, and therefore the avian community within it. If not, more detail is required about how the point count sites were selected.

4. Is the maximum number of individuals counted of each species the most suitable measure of abundance? Perhaps the mean number of each species seen per point count would be more suitable?

5. While the subsequent analytical methods of the data are generally suitable, the results are heavily reliant on the abundance estimates derived from the point counts. Therefore, more detail is required about the point count survey design and methodology.

Validity of the findings

Linked to my queries about the experimental design, although the findings are valid based on the data that the authors describe, I would like further clarification about the derivation of the species abundance estimates from the point counts as they have an important bearing on the main results.
Furthermore, the variation in detectability of the different species during point count surveys is not addressed in the methodology or discussed in the context of the findings. For example, I would assume that the smaller hummingbird species would have a much lower probability of detection than the sparrow species during point counts. Therefore, perhaps the abundance estimates from the point counts are too low for some of the species that were frequently encountered during BWC surveys. Although this is difficult to account for, it should be discussed as a limitation to the findings.
The descriptive data regarding the ages and migratory status of the different species encountered during carcass surveys are interesting, valid and discussed fully, even without the comparison to the local abundance of breeding species.
Overall the limitations presented by the described point count survey methodology, particularly related to the variation in detectability between the species, need to be discussed much more fully.

---

## Round 0.2 · accepted · Accept

Thank you for considering the changes recommended by the reviewers, including the inclusion of data from the American robins.